# Low Levels of Vitamin C during Pregnancy; a Risk Marker of Progression of Diabetic Retinopathy in Type 1 Diabetic Women?

**DOI:** 10.3390/antiox12030576

**Published:** 2023-02-25

**Authors:** Bente Juhl, Flemming Klein, Toke Bek, Line Petersen

**Affiliations:** 1Aarhus Speciallaege Clinic, 8000 Aarhus, Denmark; 2Department of Ophthalmology, Aarhus University Hospital, 8200 Aarhus, Denmark

**Keywords:** type 1 diabetes, retinopathy, pregnancy, vitamin C, ascorbic acid, NO, VEGF, cohort study

## Abstract

Pregnancy is a risk factor for the development or aggravation of diabetic retinopathy. Here, we suggest a relationship between plasma vitamin C (vitC) status during pregnancy and into postpartum in type 1 diabetes and the possible progression of diabetic retinopathy based on data of 29 women. VitC was measured in first, second, and third trimesters and three months postpartum. The women had visual acuity testing and fundus photography performed at least twice during pregnancy and onto four months after birth. An overall retinopathy grade was assigned on a scale from 0 (no retinopathy) to four according to the International Clinical Diabetic Retinopathy scale. At baseline in 1st trimester, 12 women had no retinopathy; seventeen women had retinopathy in grade 1–3. The retinopathy grade increased in nine women; remained unchanged in 17 women, and improved in three women. No women had or developed proliferative retinopathy (grade 4). The level of vitC in 1st trimester predicted the possible progression of retinopathy—the lower the vitC, the more probable the progression (*p* = 0.03; OR 1.6 (95% CI:1.06–3.2); *n* = 29 (multiple logistic regression))—while the combined levels of 1st and 2nd trimesters and the mean vitC level of the whole pregnancy did not. The diabetes duration, retinopathy grade per se in 1st trimester, 24-h blood pressure measurements, kidney function, urinary protein, HbA1c, or lipid profile were not independent predictors of progression of retinopathy during pregnancy. Retrospectively, the women who experienced progression of their retinopathy during and into postpartum had significantly lower vitC levels in 1st trimester (*p* = 0.02; *n* = 9/20), combined level of vitC in 1st and 2nd trimester (*p* = 0.032; *n* = 7/18), and mean vitC level of the whole pregnancy (*p* = 0.036; *n* = 7/9), respectively. In conclusion, our results suggest that low vitC status in pregnancy could be associated with progression of diabetic retinopathy.

## 1. Introduction

Diabetic retinopathy is one of the most common vision-threatening diseases in the western world and progression in pregnancy has been related to high levels of HbA1c, high blood pressure, low kidney function, lipid profile, longer duration of diabetes mellitus, and preexisting high level of diabetic retinopathy in [1,2,3]. However, pregnancy is also a well-known risk factor by itself of development or progression of diabetic retinopathy in women with type 1 diabetes mellitus (T1DM) [1,4,5,6,7]. The odds of progression of retinopathy in pregnant women have been reported to be about double that compared to non-pregnant women [7,8], and additionally there is a risk of progression during the postpartum [1,6,9].

While the pathophysiology of diabetic retinopathy is not fully understood, research in recent years has disclosed interesting relations between diabetic eye disease and vitamin C (VitC) metabolism. Thus, vitC plays an essential role as an antioxidant and free radical scavenger to detoxify free radicals in the retina and brain [10,11], and impaired activities of this antioxidant defense appear to be one of the possible sources of oxidative stress in diabetic retinopathy [12], leading to other key pathological processes in the development of diabetic retinopathy [13]. Since vitC has been shown to be depleted 10-fold in the vitreous humor of the eyes of patients with proliferative diabetic retinopathy compared to non-diabetic controls [14], this indicates a role of vitC in the development of diabetic retinopathy. Furthermore, in patients with diabetes mellitus, the plasma vitC levels were found to be reduced compared to normal persons [15,16,17,18,19].

While decreases during pregnancy in healthy women were found [20,21,22,23], this decrease also seems to occur in pregnant women with T1DM [23,24], and thus it can be hypothesized that the preceding low level of vitC followed by a further decrease during pregnancy might be associated with the development or progression of diabetic retinopathy. However, the relationship between vitC and diabetic retinopathy during pregnancy has not previously been examined. Therefore, in the present study, measurements of vitC and eye examinations were performed in a cohort of 29 pregnant women with T1DM during pregnancy and postpartum.

## 2. Materials and Methods

Twenty-nine pregnant women with T1DM attending the Department of Obstetrics, Aarhus University Hospital, Denmark fulfilled the inclusion criteria for participation in the study: pregestational T1DM, age > 18 years, no other systemic disease than diabetes, singleton pregnancy, and retinal examinations performed at least twice during pregnancy and onto puerperium.

The insulin treatment of the women was based on multiple daily injection insulin therapy and the women were followed at the maternity ward every other week throughout the pregnancy. Blood samples for vitC and HbA1c were taken during these routine visits as well as routine blood tests, including for kidney function. Moreover, a 24-h urine was collected for measurement of albumin excretion rate. An ambulatory 24-h blood pressure was measured using a portable oscillometry monitor in each trimester and after delivery.

Furthermore, fifteen additional women with T1DM had vitC measurements performed only in the prepregnant condition. None of these women had visual acuity testing and fundus photography performed and they served as an indicator for vitC level in the nonpregnant condition in the study. The 15 women were comparable regarding age, onset, diabetes duration, BMI and HbA1c with the present cohort of 29 pregnant women followed regarding retinopathy. Twenthyfour-h urinary albumin excretion and 24-h blood pressure were not available in this group. Nine of 15 women were non-smokers. The study was approved by the local Ethics Committee (jr.nr.1992/2523, 1998/4147 and 2026-99) and was performed in accordance with the Helsinki II declaration and all women had given their informed consent. The women were informed that the study intended to evaluate trace elements during pregnancy and not specifically the levels of vitC to avoid confounding of vitC complementary intake on the women’s own initiative on the results. The collection of samples for vitC was approved by the local Ethics Committee (jr.nr.1992/2328).

### 2.1. Ophthalmological Examination

A standard examination for diabetic retinopathy was performed at the Department of Ophthalmology, Aarhus University Hospital. Visual acuity test was performed using principles defined in the early treatment for diabetic retinopathy (ETDRS) [25] charts and mydriasis was induced by tropicamide 1% eye drops followed by fundus photography with two 60° photographs centered on the fovea and on the optic disc respectively.

The presence of each type of pathological lesion, i.e., hemorrhages and/or microaneurysms, hard exudates, cotton wool spots or vascular abnormalities, such as intraretinal microvascular abnormalities (IRMA), venous beading, or neovascularization, was noted.

Based on this, the retinopathy grade was assessed using a standardized scale: 0: no retinopathy; 1: mild non-proliferative diabetic retinopathy; 2: moderate non-proliferative diabetic retinopathy; 3: severe non-proliferative diabetic retinopathy and 4: proliferative diabetic retinopathy [26].

In each patient, the retinopathy grade on the worst eye condition according to the first available eye examination in the pregnancy was used for comparison to the last available grading for the individual women during pregnancy and onto postpartum. This was supposed to evaluate the final retinal outcome: progression, unchanged or improvement of the retinal eye status.

### 2.2. Vitamin C Measurements

Blood samples for vitC were taken in a non-fasting state to avoid hypoglycemic episodes. Blood samples for plasma vitC measurements were stabilized in sodium EDTA-anticoagulated vacutainer tubes containing dithiothreitol. Tubes were centrifuged and plasma was removed and deproteinized by the addition of 6% perchloric acid. The samples were kept at −80 °C until analysis and assayed by HPLC [27], an accepted gold standard for this measurement [28].

A plot of the ratio of vitC to internal standard vs. the concentration of 6 aqueous standards resulted in a linear curve to at least 86 μM (y = 0.16x − 0.028) R^2^ = 0.99). The within-day and day to day coefficient of variation was 2.6% and 3.9%, respectively, of a mean concentration of 19 μmol/L. The analytical recoveries were 111%, 104%, 102%, and 101% at vitC concentrations of 5.75, 28.75, 43.125, and 57.5 μmol/L, respectively.

### 2.3. Data Editing and Statistical Analysis

In the present study, we focused on the level of vitC as a marker of the possible progression of retinopathy. Thus, the predictive value of the mean of vitC samples taken in 1st trimester, the mean of the vitC in combined 1st and 2nd trimester, and of the vitC of the whole pregnancy, were evaluated regarding possible progression in retinal status during and onto postpartum (multiple logistic regression). If more than one sample of vitC per trimester was measured, the sample mean was used in the data analysis. Corresponding to this prospective analysis, a retrospective analysis of differences between two means of vitC arising from the final retinal outcome, progression or no progression, was performed in the similar group separation of vitC. This approach allowed a direct comparison of the prospective and retrospective analysis.

In other analysis, we applied the mean vitC of the single trimester and postpartum. Comparison of 2 means was performed by Student’s *t*-test if Gaussian distribution could be assured. Otherwise Mann–Whitney’s test.

We a priori calculated to have sufficient data to minimize a type 2 error (power > 80%), assuming a difference between the groups of 10 µmol/L and an expected SD of about 10 µmol/L, which was found earlier in vitC measurements of pregnancy [23].

We carried out a predefined vitC subgroup analysis using the threshold of the 50% percentile of the mean vitC of the whole pregnancy (=31.1 μmol/L) in calculating the probability for development progression or not in retinopathy (Fisher’s Exact test). The 50% percentile was an arbitrarily chosen threshold since we had no a priori research to lean on in this matter.

The time relationship between mean vitC level in each trimester and the progress of pregnancy and into postpartum were analyzed by linear regressions analysis.

The simple mean in case of more than one measurement of HbA1c, the creatinine clearance, and the measurements of urine albumin excretion rate taken in each trimester and three months postpartum were calculated and used in descriptive statistics of the cohort.

Multiple logistic regressions were used as predictive analysis of the diabetes duration, retinopathy grade per se, 24-h blood pressure measurements, urinary protein, HbA1c, and lipid profile as independently predictors of progression of retinopathy during pregnancy as the dependent variable.

Statistical analysis was performed with Sigma Plot12, Systat software.

Values are given as mean SD, unless otherwise stated. A two-sided *p* < 0.05 was chosen as level of significance.

## 3. Results

Clinical data and characteristics of the participants are shown in Table 1.

The mean HbA1c in 1st trimester and during the whole pregnancy was below 7.0% (corresponding to 53 mmol/mol) in 44% and 47% of the women respectively. HbA1c median (5–95%) of the whole pregnancy was 7.1% (5.6–9.3%). The levels of HbA1c in 1st trimester in the group that progressed in retinopathy compared to the group without progression and as a mean of the whole pregnancy, were NS: (7.5 (0.73) vs. 7.3 (0.92), *p* = 0.51) and (7.5 (0.97) vs. 7.2 (0.89), *p* = 0.32) respectively.

The levels of vitC in 1st trimester were similar whether the women had retinopathy (*n* = 17) or not (*n* = 12). The level of HbA1c, systolic and diastolic 24 h blood pressure in 1st trimester were similar whether the women had retinopathy (*n* = 17) or not (*n* = 12).

Altogether, twenty-three women had their first eye examination in the 1st trimester and, of these, one woman had her last examination in the 2nd trimester, three women had their last eye examination in third trimester, and 19 had the last examination postpartum. Six patients had the first examination in 3rd trimester and the last one postpartum.

Nineteen women were examined twice, nine women were examined three times, and one patient was examined four times in the study period from 1st trimester and onto four months postpartum.

The retinopathy progressed in nine women; remained unchanged in 17 women and improved in three women. Thus, from entry in the study to postpartum four out of 12 women with initially no retinopathy (grade 0) progressed to grade 1 retinopathy, while one progressed to grade 2; four out of 14 women with grade 1 retinopathy progressed to grade 2 while one patient returned to no retinopathy (grade 0). One out of two patients with moderate retinopathy (grade 2) improved to grade 1 while the other woman remained unchanged. One woman with severe retinopathy (grade 3) improved to grade 1. Prospectively, the level of vitC in 1st trimester predicted the possible progression of retinopathy—the lower the vitC, the more probable the progression (*p* = 0.03; OR 1.06–3.2; *n* = 29 (multiple logistic regression))—and the combined level of 1st and 2nd trimester tended to but did not reach significance (*p* = 0.06; OR 0.97–5.34; *n* = 25), while the mean vitC level of the whole pregnancy did not (*p* = NS; *n* = 16), respectively (Table 2).

We did not find diabetes duration, retinopathy grade per se, 24-h blood pressure measurements, urinary protein, HbA1c, or lipid profile to be independent predictors of progression of retinopathy, and thus these parameters were not included in the prospective multiple logistic regression analysis. We did not find smoking associated with progression of retinopathy; three out of nine smokers and six of 20 non-smokers progressed in their retinopathy respectively. The levels of vitC were similar in smokers and non-smokers (30.6 µmol/L (SD 15.0) vs. 35.7 µmol/L (SD 35.7) (pNS)) in 1st trimester.

Retrospectively, the women who experienced progression of their retinopathy during and into postpartum had significantly lower vitC levels in 1st trimester, in the combined level of vitC in 1st and 2nd trimester and in the mean vitC level of the whole pregnancy compared to the women without progression (mean (SD), *p*, *n* = +/−: 25.2 µmol/L (10.3) vs. 38.1 µmol/L (14.4), *p* = 0.02; *n* = 9/20) and (24.8 µmol/L (SD 5.86) vs. 38.7 µmol/L (SD 15.53), *p* = 0.032; *n* = 7/18) and (25.7 µmol/L (SD 7.8) vs. 40.6 µmol/L (SD 15.5), *p* = 0.036; *n* = 7/9), respectively, as shown in Table 3.

In the control group of 15 clinically comparable non-pregnant women with T1DM (Table 1), the vitC level was 40.0 µmol/L (17.8) compared to the 34.1 µmol/L (14.2) in 1st trimester indicating a lower vitC in the 1st trimester, but not statistically significant. However, a comparison between the vitC level in the non-pregnancy and in the group of women who had progression of retinopathy had a significant lower vitC in 1st trimester (25.2 µmol/L (10.3)) compared to the non-pregnant women (*p* = 0.03).

Linear regression analysis did not show a significant linear decrease of vitC levels in in relation to weeks of ongoing pregnancy, but the levels in the women with progression and no progression of their retinopathy seem to be constantly below or above throughout pregnancy and post partum respectively compared to the whole group (Figure 1).

Table 4 shows the result of Fishers exact test in the subgroups of women with a vitC level below or above the median level of the mean vitC (=31.1 μmol/L) in relation to the development or not of retinopathy.

The relative risk of development of progression was 3.75-fold higher in the group of women with a vitC level below that level than the one found in the group of women with a vitC status above this level. Relative risk was calculated as (7/14 = 0.5)/(2/15 = 0.1333) = 3.75 (see also Table 4).

In approximately 52% (15 out of 29), we observed a poor vitC status defined as a plasma vitC < 23 µmol/L at least one time during pregnancy and postpartum. In two women (7%), we found a level of <11 µmol/L postpartum corresponding to the level of scurvy. Both women had retinopathy and progressed during pregnancy.

## 4. Discussion

In the present cohort, we expected to find the level of vitC to be lower owing to the diabetic pregnancy itself than generally reported in the nonpregnant T1DM state and thereby disclosing a possible relation between vitC and the diabetic pregnancy with its “flourishing” retinopathy; a relation that is difficult to reveal when evaluating a longstanding low, but not necessarily poor level of vitamin C throughout a long term ongoing T1DM.

Seen in this perspective, the presented results seem to support our hypothesis, as discussed in the following.

**Firstly**, in the present prospective study of the course of diabetic retinopathy during/into the puerperium of 29 pregnancies, we found progression in 31% of the pregnant women; a level also reported by Phelps et al. in a similar sized study [8] and several others, as reviewed in [7], and an approximately twofold higher level of progression of no retinopathy to simplex retinopathy than in non-pregnant T1DM patient under conventional treatment [29]. The progression of retinopathy has been reported to occur at varying times during the pregnancy and our observation of worsened retinopathy also postpartum is not the most common outcome and since the women were evaluated ophthalmologically only four months postpartum, we do not know whether the progression persisted or anything about the subsequent vitC levels. However, while progression up to one year postpartum has been reported [1,9], long term studies have shown that the aggravation seems not to have long-term detrimental effects as regards the progression of retinopathy unless it has proceeded to pre-proliferative condition and phases to be followed by persistency [6].

**Secondly,** the obtained glycemic control in the present study of 7.3–7.4% was in the same range as also found in the DAPIT study of 762 women during pregnancy, and likewise the level of vitC found in our study was in accordance with the level found in the placebo group in this study; a randomized placebo-controlled trial for prevention of pre-eclampsia in women with T1DM [24]. The level of vitC in 248 women in the placebo decreased from 44 to about 35 µmol/L during pregnancy, representing a significant decrease (*p* < 0.05 calculated on the given information in the article).

We did not find a decrease of vitC in our cohort of women during pregnancy (linear regression) as found in the DAPIT study [24], but if there was any decrease of vitC, it might have occurred from the non-pregnant to the pregnant stage. The vitC level in the comparable 15 women (Table 1) with T1DM and nonpregnant was 40.0 µmol/L (17.8) compared to the 34.1 µmol/L (14.2) in 1st trimester, indicating a decrease, however not statistically significant.

In another cohort of 20 T1DM nonpregnant patient with retinopathy investigated by Juhl et al. by the same methods and the same technicians as the present study [30], the authors found a level of 50 µmol/L (36.1) compared to the level of 34.1 µmol/L (14.2) in the whole group of women in 1st trimester (Table 1). Harding et al. found a level of 43.7 µmol/L (18.4) in 735 men and women with diabetes [17].

Under all circumstances, whether the plasma vitC decreased from the non-pregnant to the pregnant condition or not, we observed a plasma vitC < 23 µmol/L in approximately 52% of the women at least one time during pregnancy and postpartum; apparently a higher proportion of women compared to approximately 25% of nonpregnant T1DM patients [30], approximately 12% in the healthy control pregnancies in the study of Juhl et al. [23], and 5% in the general population [15].

**Thirdly,** prospectively, the vitC level in 1st trimester in multiple regression analysis was an independent predictor for a progression of retinopathy; the lower the vitC, the more probable the progression of retinopathy. The combined level of 1st and 2nd trimester tended to but did not reach significance (*p* = 0.06), while the mean vitC level of the whole pregnancy did not. The significance also persisted when adjusted for common diabetic risk factors as the majority of the literature has associated it with progression of retinopathy during pregnancy [1,4,5,6,7] and this predictive ability of vitC was furthermore supported by the finding of an increased relative risk of 3.75 of progression of retinopathy if the vitC level was in the lower half as a mean of the whole pregnancy, although the latter analysis was found not significant (*p* = 0.05).

Retrospectively and in support of the level of vitC as an independent predictor for a progression of retinopathy—the lower the vitC, the more probable the progression of retinopathy—, we found that the nine women who experienced progression of their retinopathy during and into postpartum were all characterized by a significantly and constantly lower mean level of vitC (Table 3) throughout pregnancy and an even further significant decrease was observed postpartum. Thus, two (of seven) women had a level of vitC < 11µmol/L, corresponding to the level of scurvy that in the long term will lead to scurvy. Both women had progression of retinopathy during/into the puerperium four months later. Furthermore, we found that the nine women who progressed in their retinopathy had vitC levels in 1st trimester that were significantly lower than the level in the non-pregnant women (*p* < 0.03), in opposition to the levels of vitC found in the group of women that did not develop progression. This result indicates a decrease of vitC at least in the group of women with progression from the non to the pregnant stage followed with a more or less constant low vitC level just above the level of insufficiency (23 µmol/L) throughout pregnancy.

These retrospective findings indicate the importance of low vitamin C already from 1st trimester in relation to the progression of retinopathy.

The above clinical results lead to a further discussion about the relevance of the findings.

As the focus on the importance of oxidative stress in the development of diabetic retinopathy has increased, further research has disclosed interesting relations between angiogenesis in diabetic eye disease and VitC metabolism [5,6,7].

Thus, vitC has been found present in retinal and brain tissues at high concentration compared with other organs, and there is a greater than 10-fold gradient between the concentrations of vitC in the retina and brain tissues and blood [10,11].

Furthermore, the retina is reported to have the highest glucose oxidation and oxygen uptake of any tissue, thus being extremely susceptible to increased oxidative stress if activities of antioxidant defense (such as vitC and superoxide dismutase) are not present [12].

A description of the pathology associated with diabetic retinopathy is beyond the scope of this article, but recent papers have excellently reviewed the pathological features of diabetic retinopathy [13,31] based on key pathogenic processes that drive the abnormalities in the retina. In summary, the loss of vascular autoregulation leads to nutrient and oxygen deprivation of the retina, and retinal vascular BM thickening occurs and might impair cell–cell communication of the endothelium. This retinal vascular insufficiency leads to the development of diabetic retinopathy first heralded by the appearance of lesions, such as microaneurysms, maybe owing to pericyte death contributing to a weakening of the capillary. Retinal capillaries become progressively non-perfused in the diabetic retina as a direct result of vasodegeneration. The capillaries appear as naked BM tubes where the endothelial cells have disappeared secondary to pericyt death. This non-perfusion leads to ischemic pathology, with the upregulation of VEGF resulting in neovascularization and excessive vasopermeability as one the most important peptide secretions driving end stage pathology [13].

Nitrit oxide (NO) is produced in endothelial cells and exerts an important function in the autoregulation of the brain. Moreover, reduced availability of NO leads to impaired vasodilation; a well-known condition in diabetes implied in subsequent endothelial dysfunction [32,33,34,35]. VitC is directly involved in the bioavailability of NO [36,37].

An in vitro study found that ascorbic acid in cultured microvascular brain pericytes could prevent high glucose induced apotheosis [10]. In the paper, loss of pericytes is considered one of the earliest changes in the development of diabetic retinopathy associated with dysfunction of the endothelium and loss of very tight endothelial permeability barrier.

The importance of increased formation of advanced glycation end products (AGE) in diabetes was reviewed by Singh et al. The accumulation of AGEs increased the retinal endothelial cell permeability leading to vascular leakage [38]. In vitro high glucose-induced barrier leakage was mediated largely by endothelial activation of the receptor for advanced glycation end products (RAGE) since it was prevented by RAGE blockade and mimicked by RAGE ligands. VitC completely prevented RAGE ligand-induced increases in barrier permeability [39].

Another experimental study found oxidative stress to be involved in the upregulation of vascular endothelial growth factor protein in the retina in diabetic rats [40] and to increase VEGF-induced endothelial barrier permeability which was prevented by vitC [41]. As described, VEGF is considered a critical stimulus for diabetic macula edema (DME) and both retinal and choroidal neovascularization [42,43,44]. A recent in vitro study proposed vitC as a possible treatment modality in diabetic macula edema since the vitamin is severely depleted 10-fold in the vitreous humor of the eye of patients with proliferative retinopathy compared to controls [14] and on line with treatment with anti VEGF antibodies injections approved for the treatment of DME [41,45].

The key pathogenic processes discussed above offer a theoretical background for linking these events to the low vitC levels found in T1DM and even lower levels found during pregnancy as in the present study.

Limitations of the present study obviously include the small number of participants. Obtainment of vitC measurements and eye status before pregnancy in the present cohort could have contributed important predictive information on the effect of pregnancy itself on the subsequent retinopathy status and vitC levels during pregnancy in the studied cohort. Moreover, sufficient vitC measurements post-partum could have allowed for relevant evaluation in relation to the postpartum progression of the retinopathy. The samples for vitC were taken in a non-fasting state to avoid hypoglycemic episodes, which may have increased the SD of the vitC measurements and thus the risk of type 2 mistakes in the statistics.

## 5. Conclusions

In conclusion, the results from this small-sized observational study of a pregnant T1DM cohort, the first of its kind, indicate that low levels of vitC status could be associated with an increased risk of development and/or progression of retinopathy. The results are in accordance with the presented research, but further investigation is needed to verify the results. In the long term, it could have important clinical implications for T1DM pregnancy.

## Figures and Tables

**Figure 1 antioxidants-12-00576-f001:**
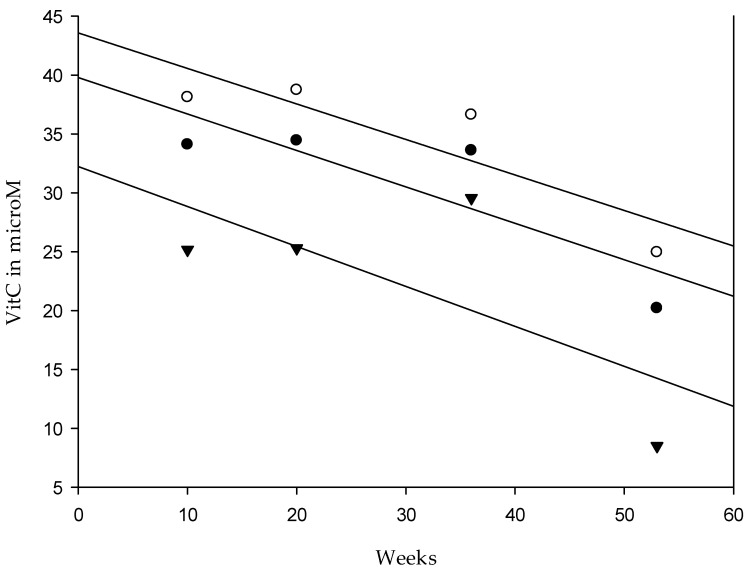
Linear regressions of mean plasma vitC levels in relations to 1st, 2 nd and 3rd trimester and postpartum in the whole group women (black dots), in the women with progression of retinopathy (black triangles) and in the women without progression (empty dots).

**Table 1 antioxidants-12-00576-t001:** Clinical data and characteristics of the study population. Descriptive statistics. Values are reported as mean (SD) for normally and as median (5–95%) for non-normally distributed quantitative data. Qualitative data as frequencies.

Variable	*n*	Value
Age (y)	29	29.3 (4.2)
Parity (*n*)	29	1.5 (1–3.45)
Weight in 1st trimester (kg)	29	64.5 (55.0–99.8)
Ht (cm)	29	167.6 (6.2)
BMI in 1st trimester (kg/m^2^)	29	24.0 (20.0–33.6)
Diabetes duration (y)	29	13.3 (8.0)
Onset (y)	29	15.9 (7.8)
HbA1c in 1st trimester (%)	29	7.3 (0.9)
HbA1c in 3rd trimester (%)	29	7.4 (1.2)
HbA1c 3 months postpartum (%)	25	8.6 (1.4)
Creatinine Clearance at entry (mL/min)	29	125.5 (24.0)
Creatinin Clearance 3rd trimester (mL/min)	29	114.0 (26.9)
Retinopathy grade 0, 1, 2, 3, 4 entry (*n*)	29	12/14/2/1/0
Retinopathy grade 0, 1, 2, 3, 4 postpartum (*n*)	29	8/15/6/0/0
Systolic blood pressure at entry (mmHg)	29	121.6 (9.3)
Diastolic blood pressure at entry (mmHg)	29	72:5 (7.0)
Systolic blood pressure 3rd trimester (mmHg)	29	125.4 (11.8)
Diastolic blood pressure 3rd trimester (mmHg)	29	74.0 (64.2–92.2)
Systolic blood pressure 3rd trimester (mmHg) #	25	122.0 (8.2)
Diastolic blood pressure 3rd trimester (mmHg) #	25	74.6 (6.5)
Urine-albumin 1st trimester (mg/24 h)	29	13.0 (4.6–721.8)
Urine-albumin 3rd trimester (mg/24 h)	29	13.5 (5–959)
Urine-albumin 3rd trimmester (mg/24 h) #	25	12.5 (5–677)
VitC in 1st trimester (µmol/L)	29	34.1 (14.5)
VitC in 2nd trimester (µmol/L)	25	30.9 (15.0–91.4)
VitC in late pregnancy (µmol/L)	16	33.6 (13.8)
VitC mean of 1st and 2nd trimester (µmol/L)	25	28.8 (16.4–73.0)
VitC mean of 1st, 2nd, 3rd trimester (µmol/L)	16	33.6 (13.2)
VitC postpartum (µmol/L)	7	10.9 (3.3–69.6)
VitC nonpregnant (µmol/L) @@	15	40.0 (17.8)
Smoker (*n* = yes/no)	29	9/20

# Four women developed preeclampsia and were not included in the statistics. The 24 h blood pressure and urine albumin excretion rate during pregnancy remained unchanged during pregnancy. @@ The 15 non-pregnant women were comparable regarding age, parity, onset, diabetes duration, BMI and HbA1c with the cohort of 29 pregnant women followed regarding retinopathy. 24-h blood pressure, and 24-h urinary-albumin excretion were not available in this group.

**Table 2 antioxidants-12-00576-t002:** Results of multiple logistic regression analysis of VitC during pregnancy in relation to the possible progression of retinopathy grade as the dependent variable from first eye examination in 1st trimester to the final eye examination.

Independent Variable	OR (95% CI)	*n*	*p* Value
VitC in 1st trimester	1.6 (1.06–3.2)	29	0.03
VitC in combined 1st and 2nd trimester	3.1 (0.97–5.3)	25	0.06
VitC in the whole pregnancy	2.2 (0.89–5.78)	16	0.08

**Table 3 antioxidants-12-00576-t003:** Results of retrospective comparison of mean vitC in the two group of women; with and without progression according to the final retinal grading.

VitC Mean(SD) µmol/L	Progression of Retinopathy	No-Progression	*p* Value
1st trimeter (*n* = 9/20)	25.2 (10.3)	38.1 (14.4)	0.023
Combined 1st and 2nd trimester (*n* = 7/18)	24.8 (5.9)	38.7 (15.6)	0.033
Mean of the whole pregnancy (*n* = 7/9)	25.7 (7.8) #	40.6 (15.5)	0.036
Postpartum (*n* = 2/5)	8.5 (3.4)	24.9 (26.7)	NS

# vitC decreased significantly from the mean level during pregnancy to postpartum in the group of women with progression of retinopathy (*p* = 0.02). Postpartum no significant difference of vitC levels was observed, whether the women had progression or not.

**Table 4 antioxidants-12-00576-t004:** Predefined vitC subgroup-analysis using the threshold of the 50% percentile of the mean vitC of the whole pregnancy (=31.1 μmol/L) in calculating the probability of the observed distribution of progression or not in retinopathy (Fisher’s Exact test).

Median vitC (31.1 µmol/L)	Progression		All Women	Fisher’s Exact Test
	Yes (*n*)	No (*n*)		
Above ≥	2	13	15	
Below <	7	7	14	
Total (*n*)	9	20	29	*p* = 0.05

## Data Availability

Data is contained within the article.

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
