# Peer review of "Low Levels of Vitamin C during Pregnancy; a Risk Marker of Progression of Diabetic Retinopathy in Type 1 Diabetic Women?"

_antioxidants, 2023, doi:10.3390/antiox12030576_

Round 1
Reviewer 1 Report (Previous Reviewer 1)
The paper is now suitable for publication
Author Response
You have proposed fine/minor spell check. This has been done.
Otherwise you found the the paper suitable for publication.
Many regards Bente Juhl MD, PhD

Reviewer 2 Report (New Reviewer)
The authors of the study undertook the task of assessing the serum concentration of vitamin C in women with type 1 diabetes in three trimesters of pregnancy and in the postpartum period. They tried to show the correlation between lower levels of vitamin C and the progression of diabetic retinopathy. The main manuscript's limitation, which was also noticed by the authors, is the small size of the research group. In addition, neither vitamin C assessment nor ophthalmologic evaluations were performed in the entire study group during the periods mentioned. Careful reading of the results allowed to detect some discrepancies between their description and the table 1. The authors wrote (quote): "4 out of 14 women with grade 1 retinopathy" (line 173) and "One out of 2 patients with moderate retinopathy (grade 2)" (line 174); while the table shows that retinopathy grade 1 initially concerned 13 pregnant women, grade 2 - 3 (quote: "Retinopathy grade 0,1,2,3,4 entry - 12/13/3/1/0").
The authors found progression of diabetic retinopathy in a total of 9 out of 29 patients, while reduction in the severity of retinopathy in 4 out of 29 women. The identical number of smokers in the study group is puzzling, according to Table 1 - 9 pregnant women were smokers. What was the level of vitamin C in this group of women and was the difference statistically significant with non-smokers? Were these women the ones with progression of retinopathy?
What do the numbers (n=9/20) ... (n=2/5) in table 3 mean?
Table 4 is completely incomprehensible - probably the configuration of the headers is in a different place than it should be.
Author Response
Please see the attachment.

Reviewer 3 Report (New Reviewer)
The paper described presumably association between low levels of vitamin C and progression of diabetic retinopathy in type 1 diabetic pregnant women. I have some concern about the paper:
1. In my opinion both title and conclusions should be corrected into: could be associated or presumably is related. The study was conducted in the small number of cases and many results are not statistically significant.
2. The results were compared only in case of vitamin C to the control group.
3. In the study group 30% of pregnant women were smoked, which can significant affect progression of the disease. In my opinion authors should include this factor into the study as well as should analyze obtained findings according to exposure to tobacco smoke. Furthermore, vitamin C could a protective function against harmful effects of smoking.
4. Why authors combined the group from 1st and 2nd trimester?
5. Why all analyzed parameters in the studied group were not assayed in the control group?
6. In my opinion characteristic of the control group should be introduced.
7. It is possible to use parametric testes, when the number of groups are so small? Because when authors used Mann-Whitney’s test the value should be represented as median value and quartile or range. Please clearly show which results were analyzed using which tests.
8. Please introduce all used methods, which were used to assay the parameters presented in Table 1.
9. Why in 2 and 3 trimester only 7 or 9 women, respectively, had assayed retinopathy grade.
10. Why the results described in verses: 160-166 are not presented in Table?
11. Table 1 two times brackets ((kg and one bracket is missed: Diastolic blood pressure 3rd trimester mmHg)#
12. Correct in all text et al. instead et all or et al (eg. verse 240, 264, 277)
13. The name of Table 3 should be moved lower (verse 198-199).
14. The list of references should be corrected, eg. position: 1, 3, 10, 29, 33, 34 or 43
Author Response
Please see the attachment.

Reviewer 4 Report (New Reviewer)
Review of the manuscript "Low levels of vitamin C during pregnancy are related to progression of diabetic retinopathy in type 1 diabetic women".
The authors should correct numerous errors in the manuscript.
Abstract.
The data in the abstract and table 1 are different.
· At baseline in 1st trimester 12 women had no retinopathy; seventeen women had retinopathy in grades 1 to 3. The retinopathy grade increased in 9 women; remained unchanged in 17 women and improved in 3 women.
· Retinopathy grade 0,1,2,3,4 entry (n) 29 12/13/3/1/0
· Retinopathy grade 0,1,2,3,4 2 trimesters (n) 7 3/2/1/1/0
· Retinopathy grade 0,1,2,3,4 3 trimesters (n) 9 2/5/2/0/0
The authors should explain.
· The level of vitC in 1st trimester predicted the possible progression of retinopathy; low vitC was the more probable progression (p=0.03; OR 1.06-3.2; n=29 (multiple logistic regression)), while the combined level of 1st and 2nd trimester and the mean vitC level of the whole pregnancy did not ((p=0.06; OR 0.97- 5.34; n=25), (p=NS; n=16) respectively.
The authors should explain: what is OR and 95% CI?
· Retrospectively, the women who experienced progression of their retinopathy during and into postpartum had a significantly lower vitC level in 1st trimester (p=0.02; n=9/20), in the combined level of vitC in 1st and 2nd trimester (p=0.032; n=7/18) and the mean vitC level of the whole pregnancy (p=0.036; n=7/9) respectively.
It is unclear how they got the number 7/18.
· In conclusion, our results suggest that low vitC status in pregnancy is associated with
progression of diabetic retinopathy.
Did pregnant women have optimal glycemic control? According to the HbA1c values, this cannot be concluded. The level of HbA1c in the first trimester of pregnancy was 7.3 (0.9) and in the third trimester 7.4 (1.2). It has been proven that hyperglycemia is the most important cause of diabetic retinopathy.
Material and Methods
· We carried out a predefined vitC subgroup analysis using the threshold of the 50% percentile of the mean vitC of the whole pregnancy (= 31.1 μmol/L) in calculating the relative risk for development progression or not in retinopathy (Fisher´s Exact test). The 50% percentile was an arbitrarily chosen threshold since we had no a priori research to lean at on this matter.
Table 4 shows the X2-test and not the relative risk.
Results
Table 1
· Parity
Nulliparous, Multiparous?
· HbA1c in 1st trimester (%) 29 7.3(0.9)
· HbA1c in 3rd trimester (%) 29 7.4(1.2)
High levels of HbA1c during pregnancy, the author needs to explain.
· Urine-albumin 1st trimester (mg/24h) 29 72.3(235.1)
· Urine-albumin 3rd trimester (m/24h) 29 102.0(306.5) high level protein conc.
· Urine-albumin 3rd trimester (24h)# 29 62.8(198.4) decrease level in 3rd trimester
Four pregnant women developed preeclampsia. Protein values in the urine decrease. It should be explained.
· VitC in 1st trimester (μmol/L) 29 34.1(14.5)
· VitC in 2nd trimester(μmol/L) 25 34.4 (19.3)
· VitC in late pregnancy (μmol/L) 16 33.6 (13.8)
There is no difference in the concentration of vitC
# Four women developed preeclampsia and were not included in the statistics, which can't be seen in table 1.
· The authors conclude: The 24-hour blood pressure and urine albumin excretion rate during pregnancy remained unchanged during pregnancy (Table 1).
· According to Table 1, HbA1c is above 7.0%. The mean HbA1c in 1st trimeter and during the whole pregnancy was below 7.0 % (corresponding to 53 mmol/mol) in 44% and 47% of the women respectively.
Table 1 shows the otherwise data.
· The authors concluded: The levels of vitC in 1st trimester were similar whether the women had retinopathy (n=17) or not (n=12). The level of HbA1c, systolic, and diastolic 24 h blood pressure in 1st trimester were similar whether the women had retinopathy (n=17) or not (n=12).
This needs explanation.
Table 3
· # vitC decreased significantly from the mean level during pregnancy to postpartum in the group of women with the progression of retinopathy (p=0.02).
Is that correct
· Figure 1. Linear regressions
Is this linear regression?
Table 3
Does Table 3 show the mean and SD or median?
· In the subgroup of women with a vitC level below the median level of the mean vitC of the whole pregnancy (31.1 μmol/L) the relative risk of development of progression was 3.75-fold higher than the one found in the group of women with a vitC status above this level (p=0.05) (Table 4).
Where relative risk is shown?
Round 2
Reviewer 2 Report (New Reviewer)
The article, which is in its current form, requires necessary graphic corrections. The data in the tables do not correspond to what they should be. The tables are completely illegible, which is probably due to the authors' desire to hastily send the article back for re-evaluation. Please also correct various typographical errors that are in the main text.
Author Response
The reviewer finds the tables illegible, as they are now, but it probably because the marking of the changes of the manuscript that has been done is present and may confuse. The data in all the tables are correct.
Many regards Bente Juhl
Reviewer 3 Report (New Reviewer)
· “These vitC measurements in the “control” group were only intended as a surrogate measure of the level of vitC in the non-pregnant condition, since we did not have paired vitC measurement in the prepregnant stage of the studied cohort of 29 women regarding retinopathy. The purpose of taken with this vitC level in the 15 women as a control group was in an effort of a discussion, whether vitC decreases or not from the non- to the pregnant and during pregnancy and not especially in relation to the other parameters in table 1 including the retinal status in the studied group. Furthermore, this part of the discussion does not change the main endpoint of the study, which concerns vitC levels only during pregnancy and into postpartum followed in relation to retinopathy”
“We think, a table with the presentation of the clinical characteristics of the control is too comprehensive in the context of its role serving only as an unpaired surrogate measure of the prepregnant vitC level. If the control group had consisted of the same women as in the pregnancy, a table would have been relevant. The discussion of the possible vitC decrease from the prepregnant to the pregnant condition is only a minor point of the purpose of the study”
Answer:
Because authors don’t want to characterized control group and they claimed that the main endpoint of the study is vitC levels only during pregnancy and postpartum followed in relation to retinopathy as well as the concentration of vitC prepregnant condition was assay in totally different group of women it is not necessary to include this group into the study.
In my opinion the investigation or the concentration of vitC is lower due to type 1 diabetes, retinopathy, pregnancy or tobacco smoke are also major factors.
· “In the present study we focused on the level of vitC as a marker of the possible progression of retinopathy. Thus, the predictive value of the mean of vitC samples taken in 1st trimester, the mean of the vitC in combined 1 st and 2end trimester and of the vitC of the whole pregnancy were evaluated regarding possible progression in retinal status during and onto postpartum. If more than one sample of vitC per trimester was measured, the sample mean was used in the data analysis”
Answer:
It is really better to see the progression of retinopathy when you add 1+ 2 trimesters as well as 1+2+3 trimester instead of each semester separately?
· “We do not find it indicated to add these references for standard tests”
Answer:
For me as a reader as well as for other Readers even short description of the methods is useful and helpful especially in the paper which not limited the number of words.
Author Response
1) The reviewer does not find it necessary to include the control group in the study, because the reviewer does not think, we have characterized the group. However, we do think, we have characterized the control group in the text and described it comparably regarding clinical data to the cohort of pregnant women. However, we have not made a table for that purpose but only written “data not shown”.
Whether or not it gives sense to include the control group is of course a matter of discussion. However, we think in lack of paired prepregnant vitC measurements of the 29 pregnant women that inclusion of the control group was the best offer to give an idea about the vitC levels before pregnancy. These 15 women were attending the same ward at the Department of Obstetrics Aarhus University Hospital in the intension to optimize the glycemic control before the women hopefully got pregnant. The statistics used is a t-test comparing unpaired data instead of statistics performed on paired data of the 29 women. The latter would of course have been the optimal.
We do agree about that the concentration of vitC is lower due to type 1 diabetes and also that retinopathy, pregnancy and tobacco smoke is related to the level of vitC.
2) The reviewer asks if it really is better to see progression of retinopathy when adding 1+2 trimester as well as 1+2+3 trimester instead of each semester separately.
We think, we have answered that question also in an earlier response to the reviewer. (Please see point 4 dated 10 February).
However elaborating the question we found it interesting to evaluate the level of vitC in the ongoing pregnancy for the single women, because it might be the level of the total vitc mean level during pregnancy that could be associated with progression of retinopathy and not only the vitc status in a single trimester. Thus it could also be the length of time with a certain vitC level e.g. low that matters for the single woman. Using this method makes it possible to account for the vitC level in the previous trimester and hereby allowing us to look at what level of vitC, the single women actually had had during pregnancy. Even, if a woman does have a high vitC level e.g. in 2. trimester, it does not exclude the possibility that a low vitc in 1. trimester might have impact on the outcome of the retinal status later in pregnancy.
3) We have been asked to check that all references are relevant to the contents of the manuscript (by the assistant editor). Thus, increasing the number of references in adding extra references regarding measurements of HbA1c (Diabetic Nephropathy 1984; 3: 92-94) and UAER (Clin.Chem. 1986; 32:1867-72) do not have a high priority.
The most important references regarding vitC measurements and retinopathy grading is given in the manuscript.
Reviewer 4 Report (New Reviewer)
the authors have answered all the reviewers' questions
I have no more comments
Author Response
Thank you for your support!
This manuscript is a resubmission of an earlier submission. The following is a list of the peer review reports and author responses from that submission.
Round 1
Reviewer 1 Report
This is a very interesting study, demonstrating a correlation between VitC levels and diabetic retinopathy progression during pregnancy in women with T1DM. This is very important for all the healthcare providers dealing with this condition and the paper deserves publication.
The paper seems scientifically sound and both the methods and the data are well presented. The discussion is supported by the data.
Before publication, however, the authors should explain why they are publishing these results 30 years after the end of the study and confirm that their technique for dosing VitC is still adequate to today standards.
Indeed, also the other paper relative to this series of patients (ref. 24, 32, 33, 34) were published 25 years after the end of the study. It seems to me that not all these citations are relevant to the present study.
Minor point: please check spelling and punctuation.
Reviewer 2 Report
Overall, the paper is old-fashioned and based on the papers published in 80 and 90 years of the previous century. In my opinion the introduction in the abstract is too long, two following sentences are not necessary „In 1938 Hanum described “diabetic retinitis” as a consequence of capillary fragility; a classic characteristic of scurvy; thus he measured plasma vitamin C (vitC) and found it low. Low vitC 9 levels in type 1 diabetes mellitus (T1DM) is well documented and recently by Juhl et al described even lower in T1DM pregnant women.”
The study is describing patients who were seen in years 1992-1994, it means that it was 20 years ago. Why it was not published earlier? The treatment of diabetes was different 20 years ago. Also, diagnostic methods in DME are complitely different nowadays (OCT). OCT examination was not performed, althought it is currently the best method to detect fluid within the retina. Diabetic macular edema (DME) may be present without hard exudates in the fundus. Very old publications have been used as the sourse for criteria of diabetic retinopathy (1991, 1984).
This study is not scientifically sound to be published in Antioxidants journal.